# Structural and Functional Insights into the Stealth Protein CpsY of *Mycobacterium tuberculosis*

**DOI:** 10.3390/biom13111611

**Published:** 2023-11-03

**Authors:** Dafeng Liu, Cai Yuan, Chenyun Guo, Mingdong Huang, Donghai Lin

**Affiliations:** 1MOE Key Laboratory of Spectrochemical Analysis & Instrumentation, Key Laboratory of Chemical Biology of Fujian Province, College of Chemistry and Chemical Engineering, Xiamen University, Xiamen 361005, China; 20520190154126@stu.xmu.edu.cn (D.L.); guochy78@xmu.edu.cn (C.G.); 2College of Biological Science and Engineering, Fuzhou University, Fuzhou 350108, China; cyuan@fzu.edu.cn; 3College of Chemistry, Fuzhou University, Fuzhou 350108, China

**Keywords:** *Mycobacterium tuberculosis*, stealth protein CpsY, crystal structure, phosphotransferase activity

## Abstract

*Mycobacterium tuberculosis* (*Mtb*) is an important and harmful intracellular pathogen that is responsible for the cause of tuberculosis (TB). *Mtb* capsular polysaccharides can misdirect the host’s immune response pathways, resulting in additional challenges in TB treatment. These capsule polysaccharides are biosynthesized by stealth proteins, including CpsY. The structure and functional mechanism of *Mtb* CpsY are not completely delineated. Here, we reported the crystal structure of CpsY^201−520^ at 1.64 Å. CpsY^201−520^ comprises three β-sheets with five α-helices on one side and three on the other. Four conserved regions (CR1–CR4) are located near and at the base of its catalytic cavity, and three spacer segments (S1–S3) surround the catalytic cavity. Site-directed mutagenesis demonstrated the strict conservation of R419 at CR3 and S1–S3 in regulating the phosphotransferase activity of CpsY^201−520^. In addition, deletion of S2 or S3 (∆S2 or ∆S3) dramatically increased the activity compared to the wild-type (WT) CpsY^201−520^. Results from molecular dynamics (MD) simulations showed that S2 and S3 are highly flexible. Our study provides new insights for the development of new vaccines and targeted immunotherapy against *Mtb.*

## 1. Introduction

Tuberculosis (TB) is a chronic communicable disease caused by infection with *Mycobacterium tuberculosis* (*Mtb*) and remains a growing threat to human health and public safety [1,2,3,4]. TB can be exacerbated in the presence of multiple comorbidities, including infection with HIV and/or SARS-CoV2, leading to high mortality [5,6,7,8,9]. More seriously, *Mtb* can evade surveillance and clearance by the host immune system and survive in macrophages for many years, posing additional challenges to TB treatment [10,11,12]. To evade the host immune system and survive, *Mtb* has skillfully evolved many strategies, such as disrupting phagosome–lysosome fusion, inhibiting phagosome acidification, suppressing apoptosis and autophagy pathways, hindering the production of interferons, inhibiting lysosomal and autophagic functions, and inhibiting the TNF-α-mediated pathway [10,11,12,13,14,15,16].

A particularly effective strategy for immune evasion by *Mtb* is the formation of an extracellular envelope composed of surface polysaccharides. *Mtb* capsule polysaccharides (CPSs) provide protection against the host immune response, thereby favoring *Mtb* survival [17,18,19,20,21]. CPSs consist of phosphodiester-linked hexa- or heptasaccharide repeating units that are synthesized by a series of transferases [22,23], including the phosphotransferase CpsY [24]. *Mtb* CPS glucans are recognized by the C-type lectin DC-SIGN (dendritic cell-specific ICAM-3-grabbing nonintegrin) *via* glucosyl residues [21,25,26,27] and are mistaken for host glycogen due to their structural similarities [27,28]. As a result, *Mtb* is endowed with non-immunogenic polysaccharides that allow it to evade detection by the host immune system [21,27,29]. Until now, the structural and mechanistic details of *Mtb* CpsY remain unknown.

Here, we determined the crystal structure of CpsY^201−520^ at 1.64 Å and found that CpsY^201−520^ is identical to the full-length CpsY in terms of their phosphotransferase activity. Based on this structure, we generated a molecular model of CpsY^201−520^ in complex with its substrate uridine diphosphate-N-acetylglucosamine (UDP-GlcNAc). Further mutation experiments revealed the inactivation and activation mechanisms mediated by the spacer segments 2 and 3 (S2 and S3). Our studies lay the foundation for understanding the mechanism by which *Mtb* evades the host immune system and provide a useful guide for designing effective vaccines or targeted immunotherapies against *Mtb*.

## 2. Materials and Methods

### 2.1. Protein Constructs, Expression, and Purification

The design and synthesis of the PCR primers for recombinant CpsY (UniProt ID P9WGD1) from *Mycobacterium tuberculosis* (*Mtb*) were conducted by Shanghai Sangon Biotechnology, China. The recombinant CpsY genes were inserted into the pET28a vector, which contains the NdeI and HindIII sites, as well as the N- and C-terminal 6xHis-tag. The PCR experiments utilized the Q5 polymerase from New England Biolabs (NEB) (Ipswich, MA, USA). The PCR procedure consisted of an initial incubation of the reaction mixture at 98 °C for 3 min, followed by 30 cycles of denaturation at 98 °C for 30 s, annealing at 65 °C for 25 s, and extension at 72 °C for 1 min. A final extension step was performed at 72 °C for 10 min. Shanghai Sangon Biotechnology (Shanghai, China) carried out DNA sequencing to confirm the correct cloning sites.

The target protein expression levels were optimized by using different induction conditions, including varying the induction time and temperature (4 h at 37 °C, 12 h at 25 °C, or 18 h at 16 °C) and different concentrations of isopropyl-β-d-thiogalactoside (IPTG) (0.1 mM, 0.5 mM, or 1.0 mM). The optimal conditions were then used for small-scale expression in 10 mL of Luria-Bertani (LB) medium. For large-scale expression, 20 mL of bacterial cells were inoculated into 1 L of LB medium containing 45 μg/mL kanamycin and incubated at 37 °C with shaking at 220 rpm until the optical density (OD) at 600 nm reached 0.6–0.8. Finally, the target protein was induced using the optimal condition.

The target protein was purified using Nickel-Nitrilotriacetic acid (Ni-NTA) (Sangon Biotechnology, Shanghai, China) affinity chromatography. Bacterial cells were suspended in 45 mL of buffer and then disrupted by 650 MPa for 3 min. The insoluble cell debris was removed by centrifugation, and the soluble fraction was applied to a Ni-NTA affinity resin. After buffer washes, the target protein was eluted with 300 mM imidazole, concentrated, and subjected to size-exclusion chromatography (SEC) (GE Healthcare, Milwaukee, WI, USA) on a Superdex 200 10/300 GL gel filtration column for full-length CpsY, or a Superdex 75 10/300 GL gel filtration column for CpsY^201−520^. Target protein purity was analyzed by sodium dodecyl sulfate–polyacrylamide gel electrophoresis (SDS-PAGE). Additionally, aggregate of the target proteins were obtained in buffer I (20 mM Tris-HCl, pH 7.4, and 500 mM NaCl), but monomers of the target proteins for buffer II (20 mM Tris-HCl, pH 7.4, 500 mM NaCl, 5 mM DTT, and 5% glycerol). Expression and purification procedures for monomeric CpsY^201−520^ variants were the same as wild-type (WT) CpsY^201−520^ using the Superdex 75 10/300 GL gel filtration column in the buffer II (20 mM Tris-HCl, pH 7.4, 500 mM NaCl, 5 mM DTT, and 5% glycerol). Protein concentration was estimated by measuring the UV absorbance at 280 nm using the Beer–Lambert law [30,31]. Protein concentration was calculated by the equation:A = ɛ·c·l
where A is the absorbance at 280 nm, ɛ is the molar extinction coefficient (unit: M^−1^cm^−1^), c is concentration (unit: M), and l is the path length (unit: cm).

### 2.2. Crystallization and Data Collection

Using vapor diffusion, crystal screening of the target protein was conducted, employing seven commercial crystal sparse matrix screen kits, including JCSG, PEG-Rx, PEG-Ion, Index, PEG-screen, Morpheus, and PEG-salt. In the initial screening stage, different protein concentrations (1.0, 2.0, and 4.0 mg/mL) were mixed with 0.5 μL of the crystallization buffer. Each well was filled with 60 μL of the corresponding crystallization buffer. Crystals were initially obtained at 25 °C. Subsequently, optimized crystals were obtained by vapor diffusion at 18 °C by combining 1 µL of protein (at a concentration of 1.4 mg/mL) with 1 µL of a reservoir solution containing 0.03 M sodium nitrate, 0.03 M sodium phosphate dibasic, 0.03 M ammonium sulfate, 0.1 M imidazole, pH 6.5, and 12% (*v*/*v*) polyethylene glycol (PEG) 4000. For data collection, these crystals were protected with 26% glycerol in the reservoir solution and then rapidly frozen in liquid nitrogen. Diffraction data were collected at beamlines BL02U1, BL19U1, and BL18U1 at the Shanghai Synchrotron Radiation Facility (SSRF) in Shanghai, China.

### 2.3. Structure Determination

All collected data were integrated and scaled in HKL2000 packages [32] and further processed in the program from the CCP4 v7.0.024 suite [33,34,35,36]. The crystal structure of CpsY^201−520^ was determined by the molecular replacement using PHASER from the CCP4 v7.0.024 suite [36] based on the full-length model predicted by AlphaFold2 (https://alphafold.ebi.ac.uk/entry/P9WGD1) URL (accessed on 3 April 2023) [37,38]. The structure was further built and refined using COOT v0.8.6.1 [39,40] and Phenix v1.13 [41,42,43]. PyMOL v2.3.4 (https://www.pymol.org/2/) URL (accessed on 23 November 2022) was also used to generate surface electrostatic potentials. The program LSQKAB from the CCP4 v7.0.024 suite [44] was used to align crystallographic structures and calculate the root mean square deviation (RMSD) on Cα atoms. Structural images were generated with PyMOL v2.3.4 (https://www.pymol.org/2/) URL (accessed on 23 November 2022), and PISA was used to calculate interfacial areas [45].

### 2.4. Enzymatic Activity Assays

Enzymatic activity assays were performed using a modified version of a previously reported method [46,47,48,49]. The optimal concentration of proteins was determined by incubation with uridine diphosphate-N-acetylglucosamine (UDP-GlcNAc). K_m_ and K_cat_ values were calculated by using the optimal protein concentrations and varying concentrations of UDP-GlcNAc or glucan. The experiment was performed by incubating 50 nM protein with 10 mM glucan, 1.5 mM UDP-GlcNAc, and 25 U/mL of Quick CIP in a buffer (20 mM Bis-Tris, pH 7.0, 50 mM NaCl, 5 mM MgCl_2_) at 37 °C for 60 min. The CIP phosphatase was used to convert UMP into phosphate and uridine. BIOMOL Green phosphate detection reagent was added and incubated for 30 min, and the optical density (OD) values were measured at 620 nm (Appendix A). The response was quantified using a phosphate standard curve. The enzymatic assays of CpsY^201−520^ variants were measured using the same protocols of wild-type (WT) CpsY^201−520^. The Michaelis constant (K_m_) and the catalytic constant (K_cat_) values were determined by using the optimal concentrations of CpsY^201−520^ and different concentrations of UDP-GlcNAc or glucan. The Michaelis constant (K_m_) and the catalytic constant (K_cat_) were determined using Hanes–Woolf plots.

The Michaelis constant (K_m_) is the substrate concentration at which the reaction rate is at half-maximum. K_m_ was calculated by the equation:V = V_max_[S]/(K_m_ + [S])
where V_max_ is the maximum rate for the catalyzed reaction, and [S] is the substrate concentration.

The catalytic constant (K_cat_) is defined as the maximum number of substrate molecules converted to product per enzyme molecule per second. K_cat_ was calculated by the equation:K_cat_ = V_max_/[E]
where V_max_ is the maximum rate for the catalyzed reaction, and [E] is the enzyme concentration.

### 2.5. Molecular Dynamics (MD) Simulations

All simulations were performed using the GROMACS 2022.3 package [50] in combination with the AMBER99SB-ILDN force field [51] and TIP3P water model. We used the Verlet cutoff scheme for neighbor searching and the Particle Mesh Ewald (PME) method [52,53] for electrostatic interaction calculation. The simulation box size was optimized to ensure a distance of greater than 1.0 nm between each atom of the protein and the box. The box was then filled with water molecules based on a density of 1 g/cm^3^. Sodium ions (Na^+^) and chloride ions (Cl^−^) were subsequently added to neutralize the simulation system and maintain a concentration of 0.15 mol/L. The energy, electrostatic, and van der Waals interactions were calculated using the Particle Mesh Ewald (PME) method. Periodic boundary conditions were applied for all simulations. The solute and solvent were separately coupled to an external temperature bath using the Nose–Hoover method [54] and a pressure bath using the Parrinello–Rahman method [55]. The temperature and pressure were maintained at 310 K and 1 bar with a coupling constant of 0.4 and 0.1 ps, respectively. The neighbor list was updated every 10 steps using a verlet buffer. Before the production run of MD simulations, all the systems were first energy minimized using the steepest algorithm, sequentially equilibrated under NVT (310 K) ensemble for 100 ps, and then under NPT (310 K and 1 bar) ensemble for 100 ps. After that, each simulation was run for 200 ns, with an integration time step of 2 fs. Unmodeled residues of the shortened CpsY^201−520^ protein were complemented based on the model predicted by AlphaFold2 (https://alphafold.ebi.ac.uk/entry/P9WGD1) URL (accessed on 3 April 2023) [37,38]. The trajectory analysis, such as the root mean square deviation (RMSD) and root mean square fluctuation (RMSF), was performed using GROMACS utilities and plotted in xmgrace.

### 2.6. Statistical Analysis

Experiments were performed at least three times, and the results are presented as mean ± SD. Statistical analysis was performed using Origin 8.5, Microsoft Excel 2013, and SPSS 19.0. Statistical significance was determined by the *p*-value; *p* < 0.05 and *p* < 0.01 were considered to be significant and highly significant, respectively.

### 2.7. Data Deposition

The atomic coordinates and structure factors of CpsY^201−520^ have been deposited in the Protein Data Bank (PDB) under the accession code 8J2N.

## 3. Results

### 3.1. Preparation and Crystallization of CpsY^201−520^

The full-length *Mtb* CpsY comprises the following regions: the N-terminal residues (aa 1–200), the C-terminal residues (aa 521–532), and the central region (aa 201–520) (Figure 1A and Appendix A) [30,31]. The central region consists of four conserved regions (CR1-CR4) separated by three spacer segments (S1–S3) (Figure 1A, Appendix A) [56,57]. To investigate the relationship between the function and structure of CpsY, our original goal was to determine the structure of the full-length protein. The full-length CpsY was expressed well in *E. coli* as a recombinant protein but appeared as aggregate in buffer (500 mM NaCl, 20 mM Tris-HCl, and pH 7.4) using size exclusion chromatography (SEC) with a superdex 200 10/300 column (Appendix A). Such aggregate was broken up by adding dithiothreitol (DTT) and glycerol to the buffer. Then, monomeric CpsY was obtained using SEC with a superdex 200 10/300 column (Appendix A).

The full-length CpsY formed crystals, but these crystals diffracted poorly and resisted further optimization (Appendix A). Subsequently, we expressed and purified CpsY^201−520^ by using the same protocols of full-length CpsY and obtained a monomeric form of CpsY^201−520^ using SEC with a superdex 75 10/300 column (Appendix A). Finally, we successfully obtained high-quality crystals of the central region (CpsY^201−520^) for structural determination (Figure 1B).

### 3.2. Determination of the Crystal Structure of CpsY^201−520^

We solved the crystal structure at 1.64 Å resolution using the molecular replacement method. The initial structure model underwent numerous iterative cycles of manual building in Coot and refinement in Phenix.refine. In the final crystal structure, the R-work and R-free values were refined to 20.3% and 22.6%, respectively (Table 1).

Two molecules are in the asymmetric unit (Appendix A). CpsY^201−520^ is composed of three roughly parallel β-sheets, consisting of two, three, and five strands, which are flanked by five α-helices on one side and three on the other side (Figure 1C and Appendix A). Additionally, four conserved regions (CR1–CR4) are located near and at the base of its catalytic cavity, with three spacer segments (S1–S3) surrounding it (Figure 1C and Appendix A). S1 was not visible in the structure due to the lack of relevant electron density, suggesting that S1 is highly flexible (Figure 1C). S2 contains a single helix, and S3 has two β-strands and two α-helices (Figure 1C and Appendix A). Three spacer segments (S1–S3) are highly flexible based on their high B-factor values obtained from crystallographic data (Appendix A). Such an arrangement may provide CpsY^201−520^ with structural flexibility to accommodate substrates.

### 3.3. Structural Comparison between CpsY^201−520^ and the Zebrafish GNPTAB

The crystal structure of CpsY^201−520^ was compared to all PDB entries using the DALI server [58], which was ranked by Z-score (Appendix A, Appendix A). The top hit in this list of entries was the zebrafish GlcNAc-1-phosphotransferase (zebrafish GNPTAB) containing uridine diphosphate-N-acetylglucosamine (UDP-GlcNAc) and Mg^2+^ (PDB code 7SJ2) with a Z-score of 29.6 (Appendix A). CpsY^201−520^ adopted an overall fold similar to that of the zebrafish GNPTAB (Figure 2), but the root mean square deviation (RMSD) value for all atoms was large (2.0 Å), and the amino acid sequence identity was only 33% (Appendix A).

CpsY^201−520^ differed from the zebrafish GNPTAB in two aspects: (1) the zebrafish GNPTAB had a long insertion in the corresponding structure of S3 of CpsY^201−520^ (Figure 2A); (2) S1 of CpsY^201−520^ possessed a flexible loop, whereas the zebrafish GNPTAB showed large α-helices in the corresponding structure (Figure 2B). These differences compelled us to further investigate the roles of the potential key residues and the spacer segments in substrate accommodation, as described in the following sections.

### 3.4. CpsY^201−520^ Accounts for the Full Catalytic Activity of Full-Length CpsY

To obtain crystallization, we excluded residues 1–200 and 521–532 and retained only residues 201–520 (CpsY^201−520^). However, the role of CpsY^201−520^ in controlling the phosphotransferase activity of the full-length CpsY remained uncertain. Therefore, we measured the phosphotransferase activities of the full-length CpsY and the truncated CpsY^201−520^ proteins, respectively (Appendix A). We found that the activity of CpsY^201−520^ was identical to that of the full-length CpsY (Figure 3A), demonstrating that CpsY^201−520^ is critical for the activity of the full-length CpsY.

The kinetic parameters of the phosphotransferase activity of CpsY and CpsY^201−520^ for glucan binding were significantly higher than those for uridine diphosphate-N-acetylglucosamine (UDP-GlcNAc) binding. Specifically, CpsY had a K_cat_ and K_m_ value of 4.4 and 31.7 times higher, respectively, whereas CpsY^201−520^ had a K_cat_ and K_m_ value of 5.0 and 16.6 times higher, respectively (Table 2).

However, the K_cat_ of CpsY^201−520^ (UDP-GlcNAc; 11.2 min^−1^) was dramatically higher than that of CpsY (UDP-GlcNAc; 4.3 min^−1^), and K_cat_ of CpsY^201−520^ (glucan; 67.5 ± 2.7 min^−1^) was also higher compared to that of CpsY (glucan; 23.3 min^−1^). On the other hand, the K_m_ value of CpsY (830.4 μM) was significantly higher than that of CpsY^201−520^ (463.6 μM) in the presence of glucan, whereas the K_m_ values for UDP-GlcNAc binding were similar for both (25.4 μM for CpsY vs. 26.3 μM for CpsY^201−520^) (Table 2).

Significantly, the second-order rate constant K_cat_/K_m_ of CpsY for glucan binding was lower than that of CpsY^201−520^, indicating that CpsY^201−520^ plays a major role in the binding of CpsY with glucan (Table 2).

### 3.5. Substrate UDP-GlcNAc Bound to CpsY^201−520^

The CpsY^201−520^ structure revealed a cavity capable of binding uridine diphosphate-N-acetylglucosamine (UDP-GlcNAc). We used homology modelling to create a structural model of the CpsY^201−520^:UDP-GlcNAc complex, using the pose of UDP-GlcNAc taken from the crystal structure of the zebrafish GNPTAB:UDP-GlcNAc complex containing Mg^2+^ (PDB code 7SJ2). The structural model showed that UDP-GlcNAc fit well into the cavity (Figure 4A,B), which had some positive charge (Figure 4A). Structurally, several key sites are located within four conserved regions (CR1–CR4) and exhibit conservation for phosphotransferase activity (Figure 4B,C; Appendix A). CpsY^201−520^ requires either magnesium or manganese for activity [46,59,60], and a magnesium ion (Mg^2+^) is coordinated by both phosphate groups and residue D324 of CR2 (Figure 4B and Appendix A). Additionally, residue R419 at CR3 in the flexible loop between helices 8 and 9 interacts with a phosphate group of UDP-GlcNAc (Figure 4B, Appendix A). On the other hand, residue E305 of CR2 in helix 5 forms hydrogen bonds with N-acetylglucosamine (GlcNAc) (Figure 4B, Appendix A). GlcNAc also forms electrostatic interactions with the side-chain atoms of residue N322 of CR2 in the flexible loop between sheets 4 and 5 (Figure 4B, Appendix A).

Mutational experiments for CpsY^201−520^ showed that mutation of R419K or R419Q abolished the phosphotransferase activity (Figure 4D), indicating that the positively charged Arg was suitably located and was stabilizing and neutralizing the negatively charged phosphate. Moreover, disruption of E305A, N322A, and D324A decreased the activity by 200- to 500-fold (Figure 4D). These results demonstrate that R419 at CR3 is essential for the phosphotransferase function of the stealth protein CpsY, which is in agreement with previous findings [46,60,61,62].

### 3.6. Conserved Regions Nearby Substrate Binding Site Affect the Activity of CpsY^201−520^

The catalytic site of CpsY^201−520^ contains three regions (aa 257–280, 201–211, and 311–321) that are close to its substrate (Figure 5A). Different recombinant *Mtb* CpsY variants were obtained in monomeric form based on size-exclusion chromatography (SEC) (Appendix A). Regions 257–260 and 267–280 are located at conserved region 2 (CR2), while 201–211 is situated at conserved region 1 (CR1) (Figure 1A,C and Figure 5B,C). When either 257–260 or 267–280 was removed (Δ257–260 or Δ267–280), or 267–280 was replaced by a 14-glycine segment (G267–280), the activity was lost (Figure 5D). However, the activity was restored when 257–260 was replaced by a 4-glycine segment (G257–260). This might be due to the increased flexibility induced by G257–260, which facilitated the accessibility of UDP-GlcNAc and/or the release of reaction products, thus aiding the catalytic process.

The restoration of the activity was observed upon replacing 201–211 with an 11-glycine segment (G201–211) (Figure 5E). Nevertheless, the deletion of 201–211 (∆201–211) resulted in a significant increase in the activity. This was because the high flexibility of 201–211 helped the binding of the substrate (UDP-GlcNAc and/or glucan) to the catalytic sites. However, when the 311–321 segment was removed (∆311–321), the activity was abolished (Figure 5F). Replacing 311–321 with an 11-glycine segment (G311–321) showed a slight decrease in activity (Figure 5F). This suggests that the flexibility of 311–321 is necessary to maintain the structural integrity of CpsY^201−520^ for the incorporation of the substrate (UDP-GlcNAc and/or glucan) into the catalytic cavity. These results indicate that these regions close to the substrate have an influence on the phosphotransferase activity of CpsY^201−520^, as their flexibility and interactions are crucial for modulating the enzymatic function and substrate binding of CpsY.

### 3.7. Deletion of Spacer Segments S2 and S3 Enhances the Catalytic Activity of CpsY^201−520^

Three spacer segments (S1–S3) are located around the periphery of the catalytic cavity of CpsY^201−520^ (Figure 6A). Variants of deleted spacer segments were respectively isolated in monomeric form by using SEC (Appendix A). Deletion of S1 (∆S1) resulted in a complete loss of phosphotransferase activity (Figure 6B), indicating that S1 is involved in the catalytic reaction. In contrast, deletion of S2 and S3 (∆S2 and ∆S3) resulted in two to four times higher activity than the wild-type (WT) CpsY^201−520^ (Figure 6B). The K_m_ value for ΔS2 or ΔS3 was dramatically lower compared to wild-type (WT) CpsY^201−520^, but the K_cat_ value was significantly higher (Appendix A), indicating that ΔS2 and ΔS3, but not WT CpsY^201−520^, have a higher affinity for the substrate and a faster reaction rate. These suggest that S2 or S3 prevents substrates (UDP-GlcNAc and/or glucan) from binding to the catalytic cavity to a certain degree, thereby inhibiting the catalytic reaction. Notably, the increase in activity was more pronounced in ∆S2 compared to ∆S3 (Figure 6B), which can be attributed to the flexible loops of S2 at both ends, resulting in more pronounced swinging motions compared to S3 (Figure 6A). These dynamic structural changes of S2 and S3 are likely to hinder substrate binding, which is supported by the B-factor values obtained for S2 and S3 (Appendix A). These results suggest that the flexibility of S2 and S3 is related to their functional role in modulating the activity of CpsY^201−520^.

On the other hand, the flexibility of each region of CpsY^201−520^ was characterized using the root mean square fluctuation (RMSF). RMSF was used to calculate the rise and fall of each atom relative to its average position, showing the structural change averaged within a range of time (Appendix A). We found that peaks of S2 and S3 were at 0.58 and 0.44 nm, respectively, which were dramatically higher compared to that of other regions of CpsY^201−520^ (Figure 7). These further demonstrated that S2 and S3 possessed high flexibility.

S2 and S3, situated around the periphery of the catalytic cavity, resemble two gates that open up the catalytic pocket. Consequently, they are important for catalytic activity, as they are likely to experience considerable structural changes in solution.

## 4. Discussion

In this work, we determined the crystal structure of CpsY^201−520^ at a resolution of 1.64 Å. CpsY^201−520^ was composed of four conserved regions (CR1–CR4) and three spacer regions (S1–S3). Results of biochemical experiments showed that the phosphotransferase activity of CpsY^201−520^ is identical to that of full-length CpsY. Mutational results demonstrated that residue R419 of CpsY^201−520^ is strictly conservative, consistent with previous reports [46,61,62]. The results of the deletion mutations demonstrated that ∆S2 or ∆S3 significantly increased the activity compared to wild-type (WT) CpsY^201−520^. We found that the S2 and S3 segments have high flexibility based on the results of molecular dynamics (MD) simulations.

Stealth proteins are characterized by four conserved regions (CRs) referred to as CR1 to CR4. The N-terminal CR1 contains a short but highly conserved sequence motif, IDVVYTF or very similar. CR2 is the most conserved member of this stealth protein family, with a length of approximately 100 residues. CR3 is less well-conserved and is about 50 residues long. The C-terminus CR4 contains an almost universally conserved tetrapeptide, CLND or CIND. In addition, divergent sequence segments of variable length, such as three spacer segments (S1–S3) of *Mtb*, are located adjacent to and between these regions. However, the roles of the three spacer segments (S1–S3) are still unknown.

Based on the structural and functional insights described above, we propose a gating regulatory model for the activation mechanism of CpsY^201−520^ (Figure 8). We suggest that CpsY^201−520^ is in flux between an active and an inactive state in solution. The two spacer segments, S2 and S3, are like two gates, regulating CpsY^201−520^ for binding to its substrate (UDP-GlcNAc and/or glucan). When both gates are closed, the catalytic pocket is contracted, preventing substrate binding, thus inactivating CpsY^201−520^. Conversely, the contact of a certain substance with CpsY^201−520^ would open both gates, stabilizing CpsY^201−520^ in an active state, which further induces conformational changes of CpsY^201−520^. The conformational changes facilitate the entry of another substrate into the catalytic pocket. Once CpsY^201−520^ and its substrates (UDP-GlcNAc and glucan) are in place, the catalytic pose is set up through structural rearrangements of CpsY^201−520^, initiating the phosphotransferase reaction. Our model not only explains our own experimental data but is also consistent with previous work reporting that the deletion of the entire S3 segment increases N-acetylglucosamine-1-phosphotransferase activity [62]. Such a gating mechanism may provide a new way to intervene with CpsY activity, which further facilitates the development of vaccines and immunotherapies for *Mtb*.

## 5. Conclusions

In this study, we have determined the crystal structure of the key fragment of CpsY (CpsY^201−520^) at very high resolution (1.64 Å). This structure unveils three β-sheets, surrounded by five α-helices on one side and three on the other. Four conserved regions (CR1-CR4) and three spacer segments (S1–S3) locate in the periphery of its catalytic cavity. We identified that R419 at CR3 and S1–S3 play critical roles in modulating the phosphotransferase activity of CpsY^201−520^. Based on our results, we outline a regulatory model for CpsY^201−520^ activation, in which the S2 and S3 serve as gates that finely control the accessibility of CpsY^201−520^ to its substrates. In the closed state, these gates cause the catalytic pocket to contract, preventing substrate binding and rendering CpsY^201−520^ inactive. Conversely, in the open state, the catalytic pocket expands, facilitating substrate binding and activating CpsY^201−520^. Our study provides mechanistic insight to Cps and will facilitate the development of vaccines and immunotherapies for *Mtb*.

## Figures and Tables

**Figure 1 biomolecules-13-01611-f001:**
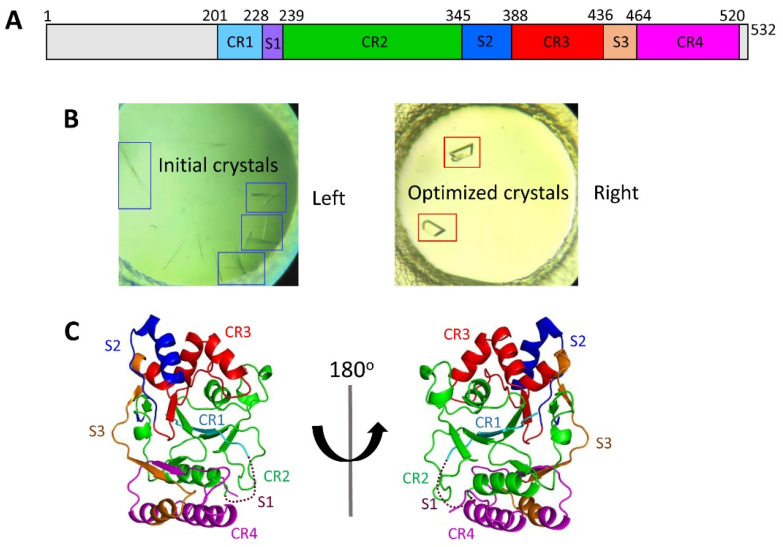
Crystal structure of CpsY^201−520^. (**A**) Schematic representation of four conserved regions (CR1–CR4, colored in cyan, green, red, and magenta) and three spacer segments (S1–S3, colored in purple, blue, and orange). (**B**) Preliminary crystals of CpsY^201−520^ were obtained by the initial screening (left, in blue frame), followed by extensive optimization, leading to diffracting crystals (right, in red frame). (**C**) Crystal structure of CpsY^201−520^ in ribbon representation, in two orientations, with the four conserved regions (CR1–CR4) and three spacer segments (S1–S3) colored, as presented in Figure 1A. Dashed lines in the structure of CpsY^201−520^ represent a loop with invisible electron densities.

**Figure 2 biomolecules-13-01611-f002:**
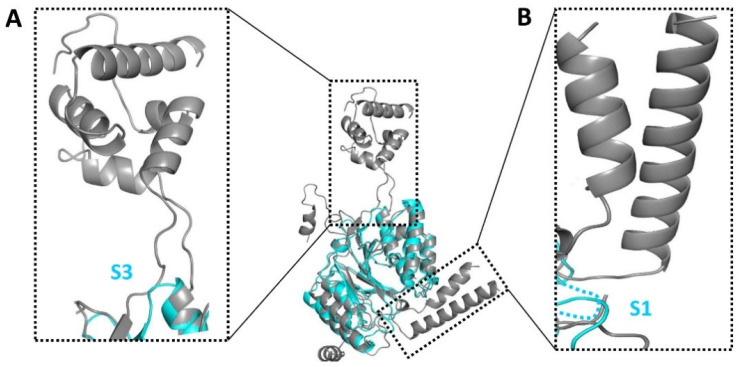
Comparative structural analysis of CpsY^201−520^ (in cyan) and zebrafish GlcNAc-1-phosphotransferase (zebrafish GNPTAB, PDB code 7SJ2, in gray). Spacer segment 3 (S3) of CpsY^201−520^ is a short loop in our structure but forms a protruding domain comprising loops and alpha helices in the zebrafish GNPTAB structure (inset (**A**)). Spacer segment 1 (S1) is disordered in our structure (shown as a dashed line) but forms a long helix (inset (**B**)) in the zebrafish GNPTAB.

**Figure 3 biomolecules-13-01611-f003:**
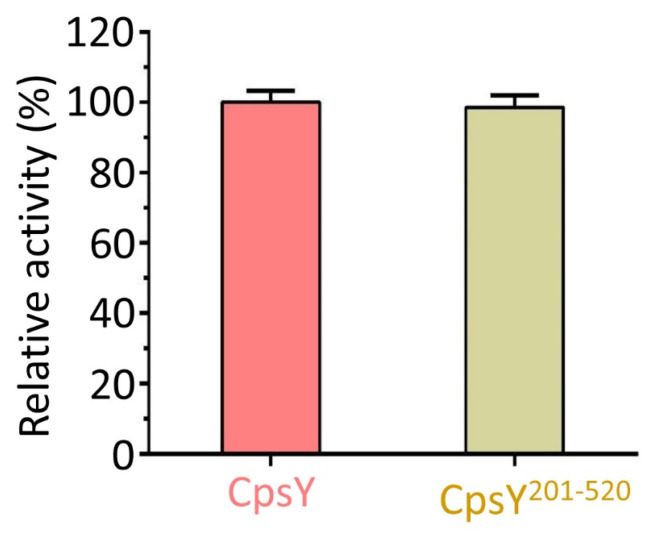
Comparative analysis of phosphotransferase activity between CpsY^201−520^ and CpsY. The phosphotransferase activity of CpsY^201−520^ was identical to that of the full-length CpsY, showing that CpsY^201−520^ was responsible for all the activity of the full-length CpsY. The activity of wild-type (WT) full-length CpsY was set to 100%.

**Figure 4 biomolecules-13-01611-f004:**
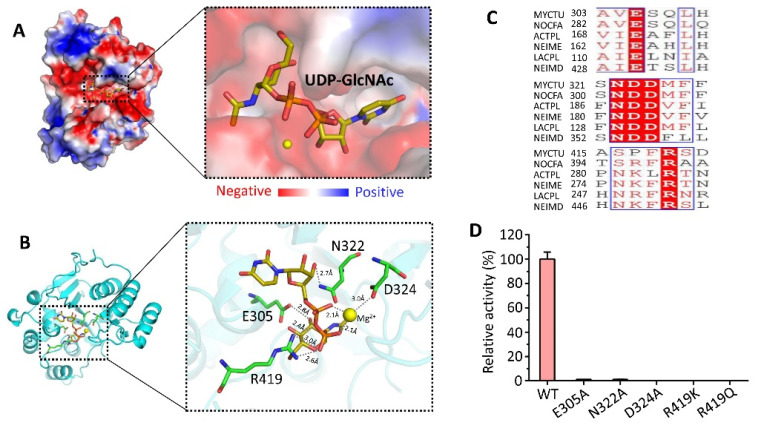
Substrate UDP-GlcNAc binds deep in CpsY^201−520^ based on the homology model of CpsY^201−520^:UDP-GlcNAc binary complex. The complex structure is shown in either surface (**A**) or ribbon (**B**) representation. The surface is colored by the electric charge: negative charges in red, positive charges in blue, and neutral charges in white. (**C**) Sequence alignment of conserved residues (in red) from various species. MYCTU, *Mycobacterium tuberculosis*; NOCFA, *Nocardia farcinica*; ACTPL, *Actinobacillus pleuropneumoniae*; NEIME, *Neisseria meningitidis*; LACPL, *Lactiplantibacillus plantarum*; NEIMD, *Neisseria meningitidis* serogroup A. (**D**) Relative phosphotransferase activities of WT CpsY^201−520^ and the indicated mutants. The E305A, N322A, or D324A mutants displayed significantly decreased phosphotransferase activity in comparison to the WT protein, whereas the R419K or R419Q mutants showed a complete loss of activity. The activity of WT CpsY^201−520^ was set to 100%.

**Figure 5 biomolecules-13-01611-f005:**
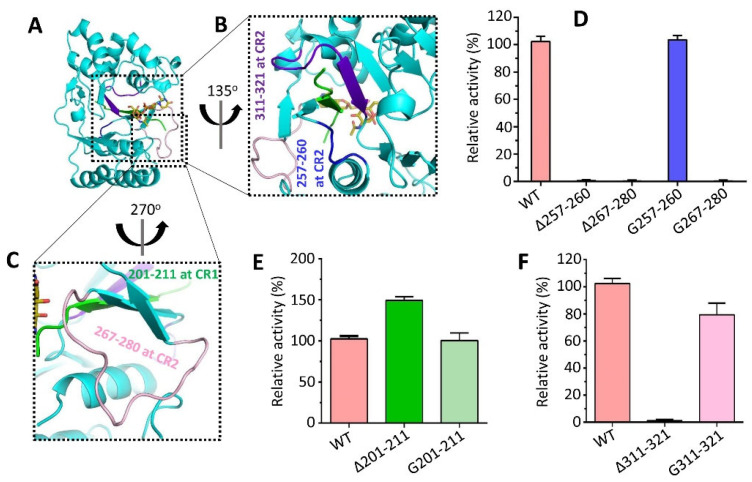
Effect of conserved regions (CRs) in CpsY^201−520^ near the UDP-GlcNAc substrate on the phosphotransferase activity of CpsY^201−520^. These regions are highlighted in (**A**–**C**): (**A**) overall structure of WT CpsY^201−520^. (**B**) The 257–260 segment is rendered in blue, and the 311–321 segment is displayed in purple-blue. (**C**) The 267–280 segment is shown in light pink, and the 201–211 segment is depicted in green. (**D**–**F**) Quantified phosphotransferase activities were observed in various variants of CpsY^201−520^ with segment deletion. The deletion of 257–260, 267–280, or 311–321 (Δ257–260, Δ267–280, or Δ311–321) completely abolished the activity, whereas the removal of 201–211 (Δ201–211) significantly increased the activity. The reference point for comparison was the 100% activity of WT CpsY^201−520^.

**Figure 6 biomolecules-13-01611-f006:**
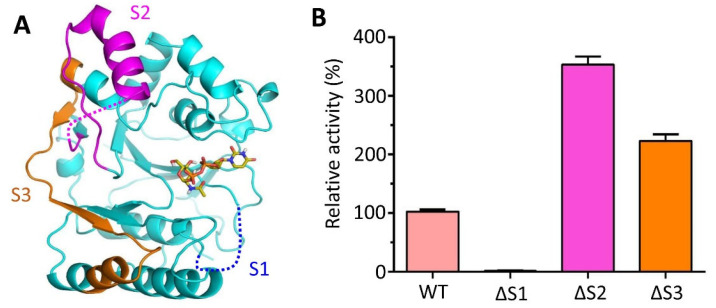
Influence of the spacer segments (S1–S3) (**A**) near the catalytic cavity on the phosphotransferase activity of CpsY^201−520^. (**B**)The right panel is the phosphotransferase activities of various variants of CpsY^201−520^ with deletions on the spacer regions. The deletion of S1 (∆S1) almost fully abolished the activity of CpsY^201−520^, while the removal of S2 or S3 (∆S2 or ∆S3) dramatically increased the activity. The reference point for comparison was the 100% activity of WT CpsY^201−520^.

**Figure 7 biomolecules-13-01611-f007:**
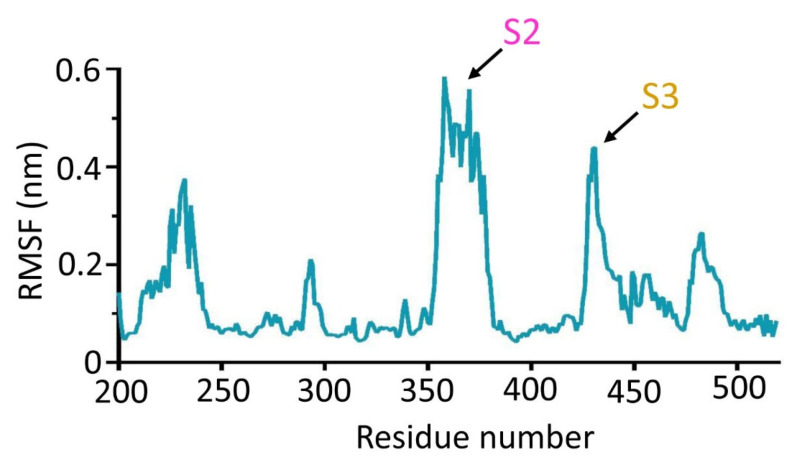
The high flexibility of S2 and S3 based on the results of the root mean square fluctuation (RMSF) profile of CpsY^201−520^. The peaks of segments S2 and S3 were significantly higher, showing that S2 and S3 possess greater volatility compared to that of other segments of CpsY^201−520^. RMSF evaluates the flexibility of a residue by measuring the relative fluctuation of atomic location in the backbone structure and determining the mean deviation of amino acid residues from a reference position in a time-bound manner.

**Figure 8 biomolecules-13-01611-f008:**
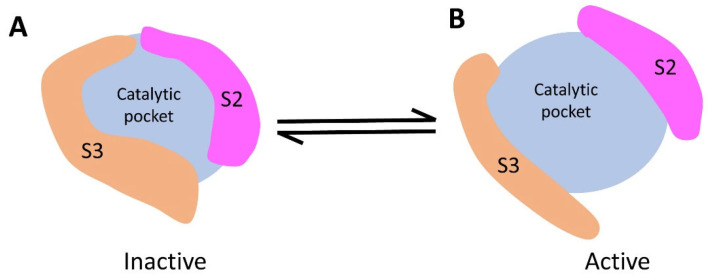
Gating regulatory model for CpsY^201−520^. Spacer region S2 (in magenta) and S3 (in orange) serve as gates regulating substrate binding. (**A**) Both S2 and S3 constrict the dimensions of the catalytic pocket, resulting in the inactivation of CpsY^201−520^. (**B**) Both S2 and S3 enlarge the dimensions of the catalytic pocket, consequently activating CpsY^201−520^.

**Table 1 biomolecules-13-01611-t001:** Data collection and refinement parameters.

	CpsY^201−520^ (PDB Code: 8J2N)
**Data Collection**	
Wavelength (Å)	0.987
Resolution range (Å)	26.10–1.64 (1.68–1.64)
Space group	P2_1_
Unit cell (a, b, c) (Å)	79.88, 41.73, 100.44
Unit cell (α, β, γ) (^o^)	90.00, 93.64, 90.00
R-merge	0.079 (1.133)
Multiplicity	6.7 (6.8)
Completeness (%)	99.9 (99.9)
Mean I/sigma(I) *^a^*	13.8 (1.7)
CC1/2	0.999 (0.720)
Total number of observations	547,798 (40,537)
Total number unique	81,599 (5998)
**Refinement**	
Resolution (Å)	26.10–1.64
Reflections used for R-free	4075 (4.99%)
R-work (%)	20.28
R-free (%)	22.62
Wilson B-factor (Å^2^)	18.9
Anisotropy	0.427
RMS lengths (Å) *^b^*	0.0066
RMS angles (^o^) *^b^*	0.852
Clashscore	1.5
Ramachandran outliers (%)	0.00
Ramachandran favored (%)	97.96
Rotamer outlier (%)	0.00
No. of atoms	4255

Values of **data collection** in parentheses are for the highest-resolution shell. *^a^* I is the mean intensity; σ(I) is the standard deviation of refection intensity I. *^b^* RMS, root mean square deviation bond length or bond angles.

**Table 2 biomolecules-13-01611-t002:** Kinetic parameters of full-length CpsY and CpsY^201−520^ constructs.

Construct	UDP-GlcNAc		Glucan	
	K_m_ (μM)	K_cat_ (min^−1^)	K_m_ (μM)	K_cat_ (min^−1^)
CpsY	25.4 ± 2.1	4.3 ± 0.4	830.4 ± 29.7	23.3 ± 1.2
CpsY^201−520^	26.3 ± 1.9	11.2 ± 0.3	463.6 ± 11.8	67.5 ± 2.7

Note: Kinetic parameters were determined using continuous phosphate detection with increasing UDP-GlcNAc concentrations or increasing glucan concentrations.

## Data Availability

The data presented in this study are available in this article (and Appendix A).

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
