# Peer review of "Structural and Functional Insights into the Stealth Protein CpsY of Mycobacterium tuberculosis"

_biomolecules, 2023, doi:10.3390/biom13111611_

Round 1
Reviewer 1 Report
Comments and Suggestions for Authors
The paper by Liu et al. "Structural and functional insights into the stealth protein CpsY of Mycobacterium tuberculosis" is devoted to structural and biochemical characterization of the stealth protein CpsY from M. tuberculosis and its mutant variants. Tuberculosis is still a serious challenge for human health and development of new strategies for combating this disease is an urgent task for molecular biology and medicine. The authors developed the protocol for bacterial production of the recombinant CpsY in the functional monomeric form and found optimal conditions for its crystallization. Crystal structure of the CpsY201-520 fragment was obtained and described. They produced and characterized several mutants of the protein and demonstrated that its truncated form is fully active. Also the significance of the conserved regions of the protein was shown by site-directed mutagenesis. The interesting part of the work was devoted to demonstration of the flexibility of the spacer regions S2 and S3 and their possible function as gate keepers. However, the described experiments are not sufficient to prove the proposed model and should be extended (see comments below). In general, the purpose of the work is clearly stated. The experiments have been carried out properly in most parts. However, their description lacks some important details (see below).
142-144 much lower and much higher in these sentences is not informative. Please provide exact values (how many times higher).
225-226 This suggests that S2 and S3 undergo structural changes that potentially prevent substrates (UDP-Glc-NAc and/or glucan) from binding to the catalytic cavity. – Deletion experiments alone are not sufficient to demonstrate structural changes of protein regions. Additional studies involving alternative biophysical methods are needed. Also, the Km and kcat values for ΔS2 and ΔS3 variants should be provided.
264-266 Through experiments of deletion mutations of S2 or S3, we found the conformational changes occurring in S2 and S3 are closely associated with the regulatory mechanism of the catalytic activity of CpsY201-520, which was further supported that the density disappearance of part of S2 and high B-factor values of S2-S3. – Described data do not provide the basis for such conclusions. Crystal structure determination and mutational analysis alone cannot demonstrate the conformational changes. Additional experiments using protein in solution should be conducted in order to support the proposed mechanism.
268 we propose a gating regulatory model for the activation mechanism of CpsY201-520 – The same comment. The proposed model is merely speculative. Instead, more detailed Discussion of CpsY structure and its comparison with homologues can be included in this Section.
Consequently, part of the Abstract which includes interpretation of these data (lines 21-24) is misleading and should be modified. Only reliable results including increased activity of the deletion mutants and RMSF data can be included.
281 Such gating mechanism may provide a new way to intervene with CpsY activity. – Please explain how this intervention could be realized.
326 The concentration and purity of the target protein was assessed using sodium dodecyl sulfate-polyacrylamide gel electrophoresis – Protein concentration cannot be reliably assessed by SDS-PAGE. Please provide the qualitative method for its determination.
368-369 The Km and Kcat values were determined by using the optimal concentrations of CpsY201-520 and different concentrations of UDP-Glc-NAc or glucan. – This is a repeat of the sentence in lines 361-363. Please describe the calculation method for these values.
Reviewer 2 Report
Comments and Suggestions for Authors
In this work, the structure of the protein exopolysaccharide phosphotransferase CpsY of Mycobacterium tuberculosis was studied. This protein belongs to the family termed Stealth, most likely involved in the synthesis of extracellular polysaccharides. Despite the fact that many proteins of this family are known in various species, the structure of exopolysaccharide phosphotransferases has not yet been experimentally resolved. In this work, the authors determined the structure of the truncated protein CpsY (from 201 to 520 amino acid residues out of 532) using X-ray diffraction analysis and conducted accompanying studies of its activity and dynamics. The article will undoubtedly attract the interest of readers. However, it requires some refinement to be able to reproduce the results.
1. Lines 62-64.
The full-length Mtb CpsY comprises the following regions: the N-terminal residues 62
(aa 1-200), the C-terminal residues (aa 521-532) and the central region (aa 201-520) (Figure 63
1A) [30, 31]. The central region consists of four conserved regions (CR1-CR4) separated by 64
It was not possible to find data on the secondary structure of proteins shown in Figure 1 in the above references [30,31].
From the text it appears that the N-terminal regions are disordered. However, the Alpha Fold model shows a different folding of the molecule, namely an ordered secondary structure of alpha helices and beta sheets. You refer to these models, but do not provide the structure:
https://www.uniprot.org/uniprotkb/P9WGD1/entry
https://www.alphafold.ebi.ac.uk/entry/AF-P9WGD1
It is not clear whether the protein structure predicted by Alpha fold is similar to the region you resolved? It would be desirable to add an overlay of these two structures to the paper to understand which part of the protein has been resolved and how similar it is to the predicted structure.
2. Lines 67-68
The full-length CpsY was expressed well in E. coli as a recombinant protein, but 67
appeared as aggregate in buffer (500 mM NaCl, 20 mM Tris-HCl and pH 7.4) using size 68
Have you tried using other types of buffers?
3. Lines 75-77 Finally, we successfully obtained 75
high quality crystals of the central region (CpsY201-520) for structural determination (Figure 76
1B). 77
What size were the crystals before and after optimization?
4. Lines 54-55
Based on this structure, we generated a molecular model of CpsY201-520 in complex with its 54
substrate UDP-GlcNAc. Further mutation experiments revealed the inactivation and acti- 55
What is UDP-GlcNAc? Please, define the acronyms when they first appear in text.
5. The crystal structure of CpsY201-520 was compared to all PDB entries using the DALI 106
server [32], which was ranked by Z-score (Table S1, Figure S5). The top hit in this list of 107
Do the similar structures found belong to stealth proteins or not?
6. Lines 139-140 Specifi- 139
cally, CpsY had a Kcat and Km value of 4.4 and 31.7 times higher, respectively, whereas 140
Please explain what the parameters Km and Kcat mean.
7. Line 168 & Lines 262-263
Mutational experiments showed that mutation of R419K or R419Q abolished the 168
tivity of CpsY201-520 is identical to that of full-length CpsY. Mutational results demonstrated 262
the strict conservation of residue R419, consistent with previous report [33, 36-38]. 263
There is no description of mutation experiments in the text. How were they carried out? The full-length or truncated proteins were expressed?
8. Lines 385-386
an integration time step of 2 fs. Unmodeled residues of CpsY201-520 were complemented 385
based on the full-length model predicted by AlphaFold2 (h?ps://alphafold.ebi.ac.uk/en- 386
Do the words "unmodeled residues" refer to the entire protein CpsY or to a shortened CpsY201-520 protein? The dynamics of which protein were simulated? Have you made a comparison of the dynamics of the full-length and truncated protein models?
The description of the molecular dynamics system is incomplete and will not allow other researchers to reproduce the simulations. The dimensions of the periodic box, the number of water molecules and ions should be added to the text of the article.
Round 2
Reviewer 1 Report
Comments and Suggestions for Authors
The authors mainly addressed the raised questions and greatly improved the text. One issue left:
lines 584-595 – It is not necessary to quote the textbook definitions of Km and Kcat. When I asked to describe the calculation method, I implied the fitting mode of the data. Was it Nonlinear Fitting function of the Origin or some other soft, or, for example, linear regression analysis from Lineweaver-Burk plot.
Author Response
The questions from the reviewers are underlined, and our answers are in plain text. The manuscript text was revised in the tracking mode of Microsoft Word.
Answers to comments of reviewer 1
(1) lines 584-595 – It is not necessary to quote the textbook definitions of Km and Kcat. When I asked to describe the calculation method, I implied the fitting mode of the data. Was it Nonlinear Fitting function of the Origin or some other soft, or, for example, linear regression analysis from Lineweaver-Burk plot.
Answer: According to the reviewer’s suggestion, we have added “The Michaelis constant (Km) and the catalytic constant (Kcat) were determined using Hanes-Woolf plots.” in the “Enzymatic activity assays” of the “Materials and methods” section.
We have also mentioned the Origin in the “Statistical analysis” of the “Materials and methods” section as follows:
Experiments were performed at least three times and the results are presented as mean ± SD. Statistical analysis was performed using Origin 8.5, Microsoft Excel 2013 and SPSS 19.0. Statistical significance was determined by the p-value: p < 0.05 and p < 0.01 were considered to be significant and highly significant, respectively.
We would like to thank the reviewers for their efforts and the professional insight they provided! We have made the revision thoroughly, and we believe the manuscript is much better now.
Reviewer 2 Report
Comments and Suggestions for Authors
Most of the issues have been resolved. I recommend the revised version of the manuscript for publication in the journal.
Lines 601-602.
distance of greater than 1.0 nm between each atom of the protein and the box. The box 601
was then filled with water molecules, based on a density of 1. Sodium ions (Na+) and chlo- 602
The units should be added after 1, i.e. 1g/cm3.
Author Response
The questions from the reviewers are underlined, and our answers are in plain text. The manuscript text was revised in the tracking mode of Microsoft Word.
Answers to comments of reviewer 2
(1) Lines 601-602.
distance of greater than 1.0 nm between each atom of the protein and the box. The box 601
was then filled with water molecules, based on a density of 1. Sodium ions (Na+) and chlo- 602
The units should be added after 1, i.e. 1g/cm3.
Answer: Thanks, we have added “g/cm3” as follows:
The box was then filled with water molecules, based on a density of 1 g/cm3.
We would like to thank the reviewers for their efforts and the professional insight they provided! We have made the revision thoroughly, and we believe the manuscript is much better now.